

# *surveyjoin*: a standardized database of scientific trawl surveys in the Northeast Pacific Ocean

Eric J. Ward[1], Philina A. English[2], Christopher N. Rooper[2],
Bridget E. Ferriss[3], Curt E. Whitmire[4], Chantel R. Wetzel[5], Lewis A.
K. Barnett[6], Sean C. Anderson[2], James T. Thorson[7], Kelli F. Johnson[5],
Julia Indivero[8] and Emily H. Markowitz[6]

[1] Conservation Biology Division, Northwest Fisheries Science Center, National Marine Fisheries Service, National Oceanic and Atmospheric Administration, Seattle, WA, United States
[2] Pacific Biological Station, Fisheries and Oceans Canada, Nanaimo, BC, Canada
[3] Resource Ecology and Ecosystem Modeling Program, Alaska Fisheries Science Center, National Marine Fisheries Service, National Oceanic and Atmospheric Administration, Seattle, WA, United States
[4] Fishery Resource Analysis and Monitoring Division, Northwest Fisheries Science Center, National Marine Fisheries Service, National Oceanic and Atmospheric Administration, Monterey, CA, United States
[5] Fishery Resource Analysis and Monitoring Division, Northwest Fisheries Science Center, National Marine Fisheries Service, National Oceanic and Atmospheric Administration, Seattle, WA, United States
[6] Resource Assessment and Conservation Engineering Division, Alaska Fisheries Science Center, National Marine Fisheries Service, National Oceanic and Atmospheric Administration, Seattle, WA, United States
[7] Resource Ecology and Fisheries Management, Alaska Fisheries Science Center, National Marine Fisheries Service, National Oceanic and Atmospheric Administration, Seattle, WA, United States
[8] School of Aquatic and Fishery Sciences, University of Washington, Seattle, WA, United States

Corresponding author
Eric J. Ward, eric.ward@noaa.gov

## ABSTRACT

Fisheries management faces challenges due to political, spatial, and ecological complexities, which are further exacerbated by variation or shifts in species distributions. Effective management depends on the ability to integrate fisheries data across political and geographic boundaries. However, such efforts may be hindered by inconsistent data formats, limited data sharing, methodological differences in sampling, and regional governance differences. To address these issues, we introduce the *surveyjoin* R package, which combines and provides public access to bottom trawl survey data collected in the Northeast Pacific Ocean by NOAA Fisheries and Fisheries and Oceans Canada. This initial database integrates over 3.3 million observations from 14 bottom trawl surveys spanning Alaska, British Columbia, Washington, Oregon, and California from the 1980s to present. This database standardizes variables such as catch-per-unit-effort (CPUE), haul data, and *in-situ* measurements of bottom temperature. We demonstrate the utility of this database through three case studies. Our first case study develops a coastwide biomass index for Pacific hake (*Merluccius productus*) using geostatistical index standardization, comparing results to independent acoustic survey estimates. The second case study examines spatial patterns in groundfish community structure, highlighting breakpoints between assemblages in their mixture of life histories and trophic compositions. Our third example applies spatially varying coefficient models to

![PeerJ]

assess sablefish (*Anoplopoma fimbria*) biomass trends, identifying regional variability in increases in occurrence and biomass. Together, these case studies demonstrate how the *surveyjoin* R package and database may improve species and ecosystem assessments by providing insights into population trends across geopolitical boundaries. This database and package represent an important step toward offering a scalable framework that can be extended to include additional data types, surveys, and species. By fostering collaboration, transparency, and data-driven decision making, *surveyjoin* supports international efforts to sustainably manage shared marine resources under dynamic environmental conditions.

## INTRODUCTION

Political and spatial boundaries pose unique challenges in fisheries management such as unintentional overfishing of stocks, uncertainties in the status of stocks, and related international conflicts and disagreements (*Song et al., 2017a*). Globally this has major financial implications for coastal countries—for those that participate in shared-stock fisheries, those stocks represent nearly half of their total catch (*Teh & Sumaila, 2015*). Spatially limited datasets (*e.g.*, datasets that do not include a species' entire range) limit our ability to track range shifts as they may not include leading and trailing edges of their distributions (*Parker et al., 2024*). Additionally, over the past decade, a growing body of research has demonstrated that changing environmental conditions are driving range shifts in species distributions into neighboring regions (*Pinsky et al., 2013*; *Maureaud et al., 2021*; *Fredston et al., 2021*). While distribution shifts may pose challenges for management, enhanced international cooperation in both scientific and management arenas may increase the likelihood of desired management outcomes in these scenarios (*Gaines et al., 2018*; *Palacios-Abrantes et al., 2020*).

Effective management of transboundary marine species relies on the ability of scientists and managers to conduct and implement population assessments across borders. However, several challenges remain, including (1) data collection protocols and standardization that differ across research institutions, leading to variability in data quality, spatial coverage, and types of observations collected, (2) limitations in data sharing rules and open science practices, (3) variation in governance priorities and management objectives, regulations, and enforcement (*Song et al., 2017b*), (4) environmental variability across regions, including spatially varying impacts of changing environments, habitat alteration, and human impacts of fisheries (*Halpern et al., 2019*; *Kwiatkowski et al., 2020*), (5) variation in socioeconomic challenges across regions (*e.g.*, economic pressures and incentives affecting coastal fishing communities and stakeholders), and (6) legal frameworks, including how fisheries management intersects with existing international agreements (*Koubrak & VanderZwaag, 2020*).

Focusing on the first challenge, recent meta-analyses using fisheries-independent survey data from around the world have highlighted the potential of standardized and accessible data (*Fredston et al., 2021*; *Maureaud et al., 2024*). Fisheries-independent data often serve as the most important data source for assessing population status through fish stock assessments. Integrating data across borders potentially allows researchers to better understand species-specific effects caused by environmental change on population status, dynamics, and distribution (*O'Leary et al., 2022*; *Parker et al., 2024*). Moreover, open and accessible data offer additional benefits, including increased trust and transparency (*Allen & Mehler, 2019*), new opportunities for research and collaboration (*Cooke & Arlinghaus, 2024*), and improved stakeholder engagement (including industry, academic partners, or non-government agencies), which collectively contribute to more efficient and informed decision making.

Though many of the previous global meta-analyses include data from government-funded fisheries surveys (*Maureaud et al., 2024*), numerous challenges remain. While valuable for global meta-analyses, due to the global focus, such datasets cannot stay as up to date or include the diversity of datatypes and surveys as required by regional assessment scientists. Assessment scientists thus end up working with multiple regional databases. As an example, survey data collected in the U.S. by the National Oceanic and Atmospheric Administration (NOAA) can be accessed *via* regional databases specific to each survey, but these regional differences result in data being provided in different formats, and data provided in different units. Currently, there is no centralized database for fish survey data within the U.S. or Canada, and this same pattern is mirrored in many other countries around the world.

With this in mind, we introduce a new publicly available database (accessible *via* an R package *surveyjoin*, https://github.com/DFO-NOAA-Pacific/surveyjoin; *Ward et al., 2025a*) that links trawl survey data collected in the Northeast Pacific Ocean, collected in Canadian waters (by Fisheries and Oceans Canada) and U.S. waters by NOAA. A previous description of this work can be described in our preprint (*Ward et al., 2025b*). For the first time, we make an effort to standardize all trawl survey data across regions and surveys. We discuss decisions for constructing the database, describe core functionality, provide three case studies to highlight the types of univariate and multivariate analyses that this new database is useful for, and discuss future directions. The resulting combined dataset will enable fishery managers, scientists, and partners to utilize fisheries resources more effectively and efficiently across international or intragovernmental borders and answer macro-scale biogeography and ecological questions.

## METHODS AND RESULTS

### Data sources

Scientists and government agencies contributing to the management of sustainable fisheries collect a wide variety of data types from a wide variety of data sources, many of which could be combined across boundaries. Because trawl surveys are one of the more common types of fisheries data collected around the world, we initially limited our focus to

creating a database of trawl survey data. Bottom trawls use nets towed on or near the ocean bottom at predefined locations and depths, and are designed to sample benthic fish and invertebrates. In many regions around the world, such surveys are used by governments and industry to monitor the sustainability of fisheries. Total catches for a particular species of interest can be converted into catch-per-unit-effort (CPUE, in kg per ha), and after accounting for sources of variability (*e.g.*, habitat, environmental, random spatial variation), data can be used to generate relative estimates of total biomass in the survey domain. Time series of relative biomass density estimates can then be used as inputs into integrated population models (*i.e.*, fisheries stock assessments).

While bottom trawl surveys have been designed to monitor temporal changes in groundfish biomass, trawl survey gear is not very selective. As a result, species encountered in these surveys include both commercial groundfish species of interest, as well as benthic invertebrates and non-commercial species. Additionally, the bottom trawl surveys are also used as platforms to collect other data critical for stock assessment and ecosystem based fisheries management, including individual lengths, weights, diet, fecundity, and age information. *In situ* oceanographic data (temperature, oxygen, salinity) may also be collected to better inform relationships between species and the environments they inhabit.

In total, *surveyjoin* brings together catch-per-unit-effort, haul, and *in-situ* bottom temperature data from 14 distinct bottom trawl surveys from across the west coast of North America (Table 1; Fig. 1). This includes five surveys in Alaska ecosystems conducted by NOAA Fisheries Alaska Fisheries Science Center since the mid-1980s (the eastern Bering Sea shelf, northern Bering Sea shelf, the eastern Bering Sea slope, the Aleutian Islands, and the Gulf of Alaska), four surveys in waters of British Columbia conducted by Fisheries and Oceans Canada since the early-2000s (Hecate Strait and Queen Charlotte Sound generally sampled in odd years, and west coasts of Haida Gwaii and Vancouver Island sampled in even years), and five surveys on the U.S. West Coast conducted by the NOAA Fisheries Northwest Fisheries Science Center. A full list of these surveys may be accessed in *surveyjoin* with the function `get_survey_names()`.

All data contained in this database and retrievable through the *surveyjoin* R package are the final, validated survey data that are publicly accessible soon after surveys are completed. Each survey listed in Table 1 collects station-level catch and CPUE, haul, and *in situ* environmental data. Raw data from each organization are also available from different sources; data collected by NOAA Fisheries' Alaska Fisheries Science Center are available through the Fisheries One Stop Shop (FOSS) platform (https://www.fisheries.noaa.gov/foss/f?p=215:200), data collected by NOAA Fisheries' Northwest Fisheries Science Center are available online (https://www.webapps.nwfsc.noaa.gov/data/map), and data collected by Fisheries and Oceans Canada is available online (https://open.canada.ca/data/en/dataset/a278d1af-d567-4964-a109-ae1e84cbd24a/resource/9992b5d6-fa0d-4dd5-a207-dd1c20ee3f6d).

While the gear and sampling protocols are largely consistent across these surveys, some differences in gear design and sampling protocols exist, which may influence catchability and selectivity (Table 1). As an example, the eastern and northern Bering Sea Shelf surveys use a smaller footrope and slightly different net design compared to other surveys in

**Table 1 Summary of trawl survey datasets included in surveyjoin.** The *surveyjoin* package brings together data from 14 distinct bottom trawl surveys from across the west coast of North America.

| Region/Organization | Survey | Years | Key differences | Most current citation |
|---|---|---|---|---|
| NOAA Fisheries, Alaska Fisheries Science Center, Groundfish Assessment Program: https://www.fisheries.noaa.gov/alaska/science-data/groundfish-assessment-program-bottom-trawl-surveys Data source: https://www.fisheries.noaa.gov/foss/f?p=215:200 | Aleutian Islands bottom trawl survey | Biennial in May–July; 1991–Present | Stratified random survey design; Poly Nor'Eastern, four-seam, hard bottom, high-rise bottom | *Von Szalay et al. (2023)* |
| | Gulf of Alaska bottom trawl survey | Biennial in May–July; 1990–Present | | *Siple et al. (2024)* |
| | Eastern bering sea crab/Groundfish bottom trawl survey | Annual in May–July; 1982–Present | Stratified systematic sampling survey design with fixed stations at center of 20 × 20 nm grid; 83–112 eastern otter trawl with wire footrope and no roller gear | *Zacher et al. (2023)*, *Markowitz et al. (2024)* |
| | Northern bering sea crab/Groundfish Survey-Eastern bering sea shelf survey extension | Intermittently in May–July; 2010–Present | | *Zacher et al. (2023)*, *Markowitz et al. (2024)* |
| | Eastern bering sea slope bottom trawl survey | Intermittently in May–July; 2002–2016 | Modified index-stratified random of successful stations survey design; Poly Nor'eastern net with mud sweep | *Hoff (2016)* |
| Fisheries and Oceans Canada, Pacific Biological Station Data source: *Cornthwaite, (2020a, 2020b, 2020c, 2020d)* | West coast of Haida Gwaii | September; 2006, 2007, and 2008–2022 in even years | Random depth-stratified surveys with 20 min tows; Atlantic Western IIA box trawl net with a 16 inch rockhopper footrope and 1,000 kg Thyboron Type II doors using chartered and government vessels | *Sinclair et al. (2003)*, *Nottingham et al. (2017)*, *Williams et al. (2018a, 2018b)*, *Wyeth et al. (2018)*, *Anderson, Keppel & Edwards (2019)* |
| | Hecate strait | May–June; 2005–2023 in odd years | | |
| | Queen Charlotte sound | July–August; 2003, 2004 and 2005–2023 in odd years | | |
| | West coast of Vancouver Island | May–June; 2004–2018 in even years, 2021, 2022 | | |
| NOAA Fisheries, Northwest Fisheries Science Center Data source: FRAM Data Warehouse: https://www.webapps.nwfsc.noaa.gov/data/map | Triennial shelf | 1980–2004 | Transects selected perpendicular to the coast, California to British Columbia | *Keller, Wallace & Methot (2017)* |
| | NWFSC slope | 1998–2002 | Fixed east–west track lines with some randomly sampled stations, U.S. west coast | |
| | NWFSC shelf | 2001 | Limited depth range (55–183 m), combined with the slope survey in later years | |
| | NWFSC hypoxia | 2007–2011 | Extension of NWFSC Combo survey to monitor dissolved oxygen | |
| | NWFSC combo | 2003–present | Random stratified sampling design using chartered fishing vessels | |

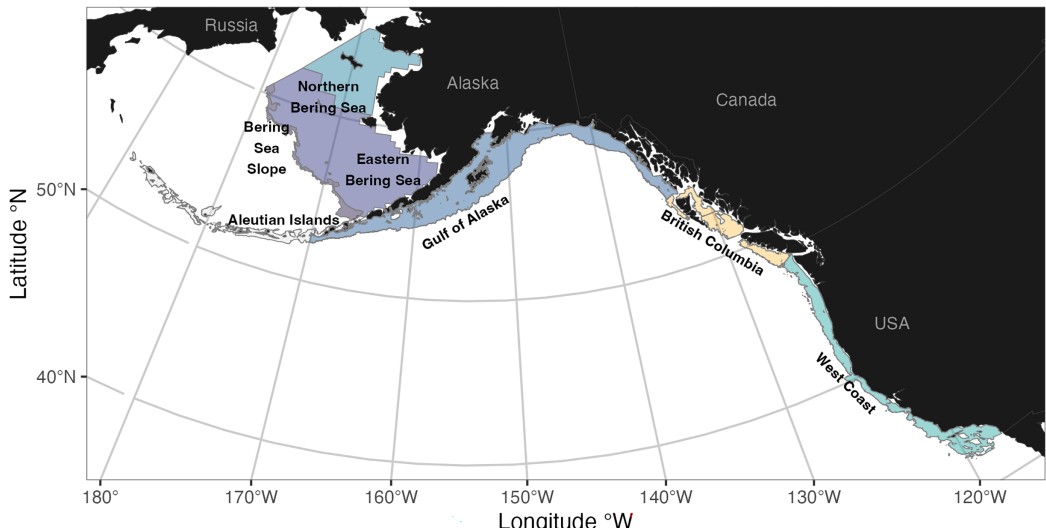

**Figure 1 Spatial coverage of bottom trawl surveys included within the *surveyjoin* R package.** Maps are generated using data from the maps R package (*Becker et al., 2025*) and data from individual surveys (Table 1).

Alaska (*Stauffer GD (compiler), 2004*). However, ongoing research to calibrate catch data between the Bering shelf and slope surveys aims to standardize catches for some common species; this region faces increased challenges because it is experiencing borealization (*Litzow et al., 2024*) and surveys are having to adapt to changing environmental conditions and species distributions (*Vilas et al., 2024*). In the absence of such calibrations or survey redesign, catchability may also be estimated statistically based on spatiotemporal proximity (*e.g.*, by including survey as a factor predictor).

Throughout the history of these surveys, there have been minimal changes in the protocols regarding gear construction, configuration, towing speeds, and survey design within an individual survey (*Vilas et al., 2024*). Without changing these concepts, technological advances (such as the development of net mensuration equipment, on-bottom sensors, and satellite-based global positioning systems) have enhanced the accuracy of effort measurement. Moreover, changes in species identification and categorization protocols, although not universally applied across all surveys, have also influenced the data collection process (*Stevenson & Hoff, 2009*).

## Package development

While each of the individual surveys in our dataset collects hundreds (or thousands) of species, we only included a subset in our initial version of the *surveyjoin* package. Each of the surveys (Table 1) can be assigned to one of four major geographical areas: Eastern Bering Sea, Gulf of Alaska, British Columbia, and west coast of California/Oregon/Washington states. Within each of the four regions, we first filtered and identified species that occurred in more than 5% of tows. In the spirit of cross-border data standardization, we then filtered that list to identify those occurring in two or more of our four geographic areas, and finally queried all data from those species across all regions. These steps resulted

**Table 2 Summary of variable names in data returned by the *surveyjoin* package.** Subset of variables included in dataframes returned by the *surveyjoin* package. This table includes the variable name and a brief description; more detailed metadata is included in the package documentation, and with the *surveyjoin* function 'get_metadata()'.

| Variable | Description |
| --- | --- |
| survey_name | Name corresponding to surveys described in Table 1 |
| event_id | Unique haul identifier |
| date | Date, in YYYY-MM-DD format |
| lat_start, lon_start, lat_end, lon_end | Positions describing the start and ending location of each haul, recorded in decimal degrees |
| depth_m | Net depth in meters |
| effort | Effort in hectares (1 km$^2$ = 100 ha) |
| bottom_temp_c | *In situ* bottom temperature, degrees Celsius |
| region | Organization responsible for collecting data (Table 1) |
| itis | Integrated Taxonomic Information System (ITIS) identifier |
| catch_numbers | Number of individual fish caught |
| catch_weight | Total biomass of fish caught (kg) |
| scientific_name | Scientific name (genus and species) |
| common_name | Common scientific name, standardized across surveys |

in nearly 3.3 million observations of 55 species. To help future efforts in using these data, we include the common name, scientific name, and Integrated Taxonomic Information System (ITIS; https://itis.gov/) identification for each species (Table 2); these may be accessed in *surveyjoin* with the function `get_species()`.

Trawl datasets were standardized across surveys by converting the raw data from each survey into the same units (effort in hectares, catch count in numbers, and weight in kilograms). In addition to catch and effort, we identified common variables across surveys often used in analyses of these data including location (start and end longitude and latitude in decimal degrees), date, and the *in situ* bottom temperature associated with each haul. Finally, because many analyses using the *surveyjoin* database will be focused on the estimation of population trends, we also include the design grids associated with each of the surveys as data objects. Each of these objects are included in the package as a dataframe (each grid consisting of >10,000 rows), with cell centroids and cell areas. Examples of using these grids with model predictions are included in the case studies below. In developing the *surveyjoin* R package, we are following best practices in code development on GitHub (*e.g.*, versioning, releases, code review), and to facilitate interpretation across regions and surveys, we link to survey-specific metadata. As our data originates from multiple agencies and regions, we are also extending these practices to versioning individual datasets. Tracking data source versions should facilitate transparency and reproducibility for end users, and allow changes to be compared over time.

## Case studies

To demonstrate the utility of the *surveyjoin* package, we developed three illustrative case studies focused on different statistical models and questions. Each of these models can be viewed as a different type of geostatistical model, where raw observations are associated

with a unique haul identifier and latitude and longitude coordinates. Data and code to replicate our analysis and figures is on our GitHub repository (https://github.com/DFO-NOAA-Pacific/surveyjoin-paper).

## Case study 1

For a first case study, a geospatial index standardization (*Thorson et al., 2015*) was applied across international borders to develop a coastwide index of Pacific hake (*Merluccius productus*, also known as Pacific whiting). Pacific hake are a fast growing, semi-pelagic species found in the northeast Pacific Ocean off the coasts of Canada and the U.S. (ranging from Alaska to California) and, by volume, represent one of the largest fisheries on the west coast of North America. Pacific hake are managed under the bilateral Pacific Whiting Agreement and an estimate of their population status is updated annually by a team of scientists from NOAA Fisheries and Fisheries and Oceans Canada. The assessment model implemented for Pacific hake can be described as a Bayesian age-structured population model based on the Stock Synthesis modeling framework (*Methot & Wetzel, 2013*), with data inputs including absolute indices of abundance from an acoustic survey (*Grandin et al., 2024*). Similar indices of hake abundance have not yet been developed from trawl surveys, as Pacific hake are semi-pelagic and may not be well sampled by bottom trawl gear (most fish sampled by trawl gear are mature, through the survey does occasionally catch immature individuals). However, information from bottom trawl gear may help provide a clearer picture of population size for the semi-pelagic species, as each survey samples different vertical portions of the water column (*Monnahan et al., 2021*). Our objectives in this analysis are to (1) construct an index using trawl data from the *surveyjoin* package and compare it to the acoustic survey, and (2) demonstrate how indices may be used to evaluate distributional shifts.

The *surveyjoin* package was used to query all haul and catch data from British Columbia and the West Coast of the U.S. from 2003 to 2023. Data from surveys in the Gulf of Alaska were not included because observations of Pacific hake are sparse. Since 2003, there have been an average of 673 hauls per year in the Gulf of Alaska, with an average of 23.5 occurrences of hake per year (ranging from 1 to 69). Bottom trawl surveys along the West Coast of the U.S. have been completed annually, except in 2020 due to the COVID-19 pandemic (*Keller, Wallace & Methot, 2017*). Surveys in British Columbia are stratified into four regions, with two regions usually sampled in odd years (Hecate Strait and Queen Charlotte Sound) and two in even years (West Coast Vancouver Island and West Coast Haida Gwaii) as discussed above (also see *Sinclair et al., 2003*; *Anderson, Keppel & Edwards, 2019*). For this analysis, we used a spatiotemporal Generalized Linear Mixed Model (GLMM) using the *sdmTMB* R package (*Anderson et al., 2025*; *R Core Team, 2024*). The *sdmTMB* package combines the Stochastic Partial Differential Equation (SPDE; *Lindgren, Rue & Lindström, 2011*; *Lindgren & Rue, 2015*) approach with fast maximum likelihood estimation using Template Model Builder (*Kristensen et al., 2016*). Given the zero-inflated and highly skewed catch data relative to most distribution families, CPUE was modeled with a delta-Gamma model (*Pennington, 1983*). Presence-absence was modeled using a Bernoulli distribution and positive catch rate as a Gamma distribution.

For both the presence-absence and positive model components, the general form of the spatiotemporal GLMM be expressed as

$$u_t = f^{-1}(Xb + \boldsymbol{\omega} + \boldsymbol{\epsilon}_t)$$

where $\boldsymbol{u}_t$ represents the predicted occurrence or density in link space at all locations $s$ in time $t$, $f^{-1}()$ is the inverse link function (*e.g.*, logit or log), $\mathbf{X}$ represents a matrix of fixed-effects coefficients (year effects) with estimated coefficients $\mathbf{b}$. We separate the spatial variation $\boldsymbol{\omega} \sim MVN(0, \Sigma_\omega)$ from the year-to-year spatiotemporal variation $\boldsymbol{\epsilon}_t$, where the spatial component represents a spatial intercept (treated as a Gaussian Markov random field) and the spatiotemporal component represents spatially correlated temporal deviations from $\boldsymbol{\omega}$. Because of the even-odd year sampling and checkerboard pattern in the Canadian survey data, the spatiotemporal fields were modeled as an AR(1) process. Year effects were included as predictors (factors) in both model components, and did not include other predictors. However, exploratory versions of the model that included survey catchability effects resulted in models not converging. Future work could consider gear-level catchability effects (*e.g.*, *Davidson et al., 2025*) or priors on catchability effects (*e.g.*, *Monnahan et al., 2021*). Convergence was assessed by assuring that the Hessian matrix was positive definite and all absolute log likelihood gradients were <0.001. Residuals were checked *via* the DHARMa package (*Hartig, 2024*).

After fitting a spatiotemporal model, the second step in geostatistical index standardization is making predictions to a gridded surface and computing a weighted sum of predicted biomass across grid cells by year (*Thorson et al., 2015*). Because our model fitting process was restricted to surveys from British Columbia and the West Coast of the U.S., we used the combined survey grids from these regions for the prediction grid. The "epsilon" bias correction in sdmTMB was implemented, which accounts for non-linear transformations when dealing with random effects (*Thorson & Kristensen, 2016*). Estimated hake biomass time series indicates that the population declined over the 2003–2015 period, and has since increased slightly (Fig. 2). In contrast, the acoustic survey over the same period is variable and without any clear trends. The same prediction grid used to generate coastwide biomass indices may be further subdivided to quantify regional trends. As an example, we generated three regional indices, representing biomass in Canadian waters and biomass north and south of 42° latitude (this breakpoint corresponds to the northern border of California, but also represents a biogeographic break in the California Current; *Sivasundar & Palumbi, 2010*). Comparing recent trends in these indices suggests that over the last 5 years, biomass in Canada has remained relatively unchanged while biomass in the U.S. has increased slightly (Fig. 2).

## Case study 2

In our second case study, we used data from the *surveyjoin* package to characterize variation in groundfish community structure among species with overlapping ranges. The diversity and composition of groundfish communities are known to be structured by oceanography (*e.g.*, currents), bathymetry (*e.g.*, shelf width), environmental conditions (*e.g.*, temperature), habitat types (*e.g.*, substrate), and fishing pressure

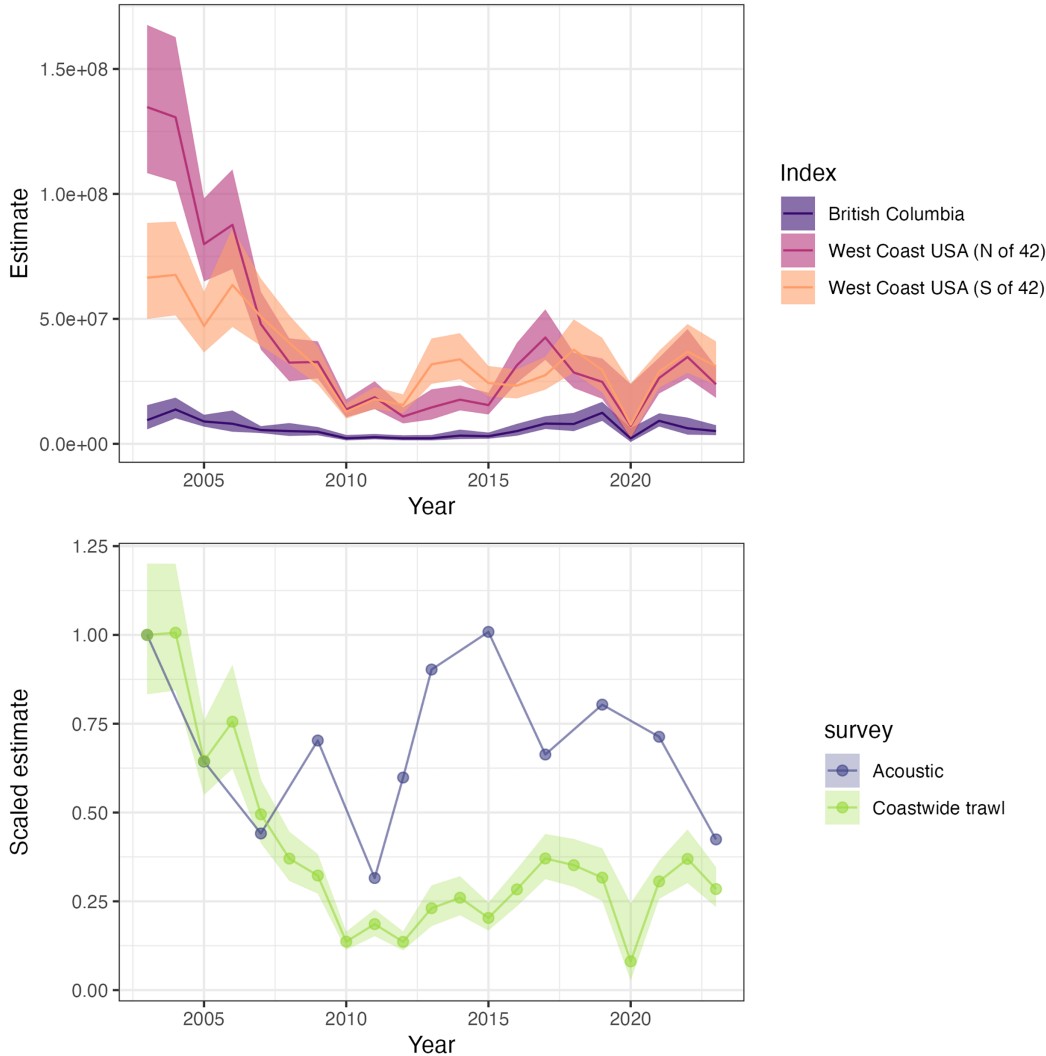

**Figure 2 Estimated biomass trends for Pacific hake on the west coast of the U.S. and Canada.**
Estimated relative biomass (with 95% confidence intervals as ribbons) trends for Pacific hake (*Merluccius productus*) using trawl survey data from the west coast of the U.S. (California, Oregon, Washington) and Canada (British Columbia). The top panel illustrates trends for British Columbia, and north and south of 42° latitude (S of 42° corresponds to California, N of 42° corresponds to Oregon and Washington state). The bottom panel shows a comparison between the aggregate coastwide index from the trawl survey *vs* the acoustic biomass estimate used for the international stock assessment (both series are scaled by their respective mean value in 2003 to place them on similar scales; confidence intervals are only available for the trawl survey index).

(*Howard et al., 2021*). Understanding how groundfish communities vary spatially can add to our understanding of these dynamics, as well as the resulting influences on food web dynamics and connectivity of stocks among management regions (*e.g.*, across national borders). The ability to describe groundfish communities across this broad spatial range also facilitates the creation of indicators to rapidly detect community shifts in the future, providing an early warning tool for fishery managers and members of the fishing industry

to adapt to environmental change (*Thorson, Pinsky & Ward, 2016*; *Fredston et al., 2021*; *Thompson et al., 2023*).

The *surveyjoin* package was first used to query survey data from 2003 to 2023 for a subset of groundfish with similar ranges across the northeast Pacific that are well-sampled by the bottom trawl surveys. Species included arrowtooth flounder (*Atheresthes stomias*), Pacific halibut (*Hippoglossus stenolepis*), flathead sole (*Hippoglossoides elassodon*), Pacific cod (*Gadus macrocephalus*), rex sole (*Glyptocephalus zachirus*), Pacific ocean perch (*Sebastes alutus*), sablefish (*Anoplopoma fimbria*), Dover sole (*Microstomus pacificus*), English sole (*Parophrys vetulus*), and lingcod (*Ophiodon elongatus*).

To understand the variation in groundfish community structure, we constructed a joint (multispecies) species distribution model that included an intercept and an independent spatial random field for each species (with shared range or decorrelation parameter, controlling the distance at which two points are functionally independent). We used an isotropic Matérn correlation function defined by spatial coordinates in latitude-longitude, using a 0.5 degree cutoff to define the SPDE mesh. A Tweedie distribution (log link) was used to represent observation error for each species. Model fitting was done using the *tinyVAST* package (*Thorson et al., 2025*). After evaluating convergence and diagnostics, as in Case Study 1, we used the fitted model to predict population density for each species across the survey domain.

Using these predictions of population density, we performed a cluster analysis to summarize variation in species' biomass across the domain and illustrate how community structure varies over space. Ward's dissimilarity for log-density between each pair of locations (*fastcluster* R package; *Müllner, 2013*) was calculated, and then hierarchical clustering was used to identify clusters. Using a value of K = 3 clusters, we then calculated the average log-density for each species.

Cluster analyses identified breakpoints between the three groundfish community components across the domain: (1) near Cape Mendocino, California, in the south and (2) the intersection of the Alaska peninsula and the Aleutian Islands in the north (Fig. 3). The southern community cluster was characterized by relatively higher biomass estimates of Dover sole, sablefish, rex sole, English sole, and lingcod. The central community ranging from Washington state north through British Columbia and the Gulf of Alaska is characterized by relatively higher biomass estimates of arrowtooth flounder, Pacific halibut, sablefish, and Pacific cod. The third community, located in the Aleutian Islands, had relatively equal representation across species, with slightly lower abundances of the southern species (*e.g.*, English sole, lingcod). Flatfish species clustered together by similar life-history traits, including small-mouthed flatfishes (English sole, rex sole, and Dover sole) and larger piscivorous flatfish (arrowtooth flounder and Pacific halibut). Some of the patterns among species can be explained by ontogenetic habitat shifts, such as sablefish moving to the edge of the continental shelf down the continental slope as they mature, thus becoming less available to the regional surveys that are restricted to the shelf (*e.g.*, in the Aleutian Islands and western Gulf of Alaska).

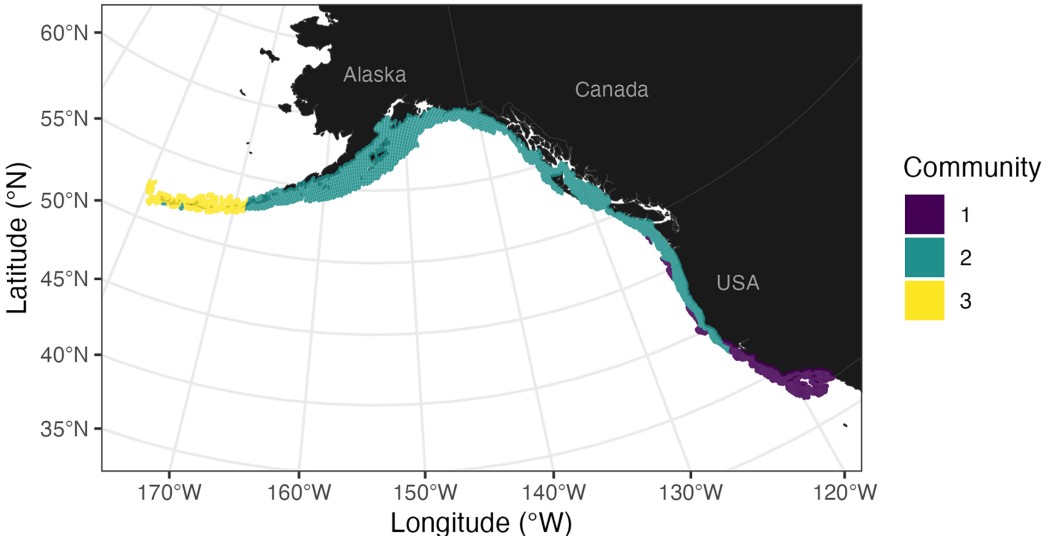

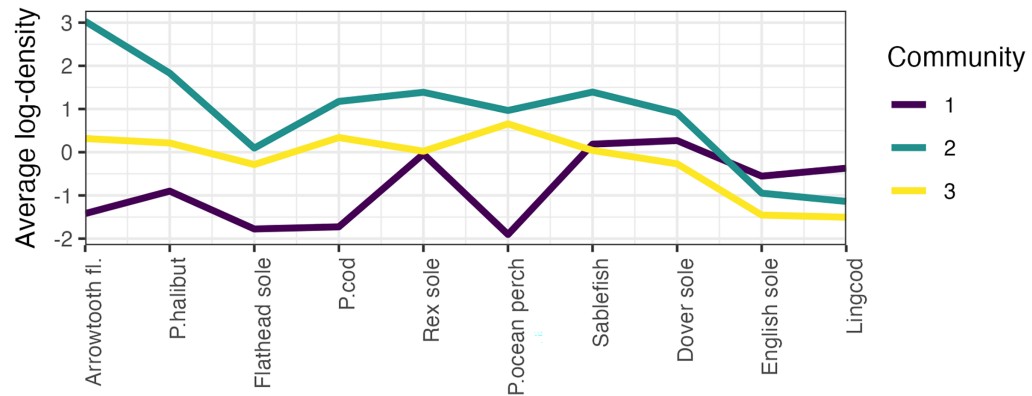

**Figure 3 Multispecies trends in the U.S. and Canada estimated from trawl survey data.** Results from multispecies analyses of joint bottom trawl survey biomass for arrowtooth flounder (Arrowtooth fl.), Pacific halibut (P. halibut), flathead sole, Pacific cod (P. cod), rex sole, Pacific ocean perch (P. ocean perch), sablefish, Dover sole, English sole, and lingcod, in the shelf waters of Alaska, British Columbian, and the west coast of the U.S. (California, Oregon, Washington). The top panel illustrates the spatial distribution of each cluster; the bottom plot identifies species' representation within each cluster by average log-density. Maps are generated using data from the maps R package (*Becker et al., 2025*) and data from individual surveys (Table 1).               

## Case study 3

For our third case study, we demonstrated how trawl survey datasets can be input into a coastwide model to quantify changes in time and space at fine spatial scales. This case study focused on sablefish, which is one of the most commercially important species in the northeast Pacific and for which there has been recent interest in integrating science and management coastwide (*Kapur et al., 2024*). Models with spatial covariates or trends are becoming increasingly used in fisheries and ecology, and are also referred to as spatially

varying coefficient (SVC) models (*Hastie & Tibshirani, 1993*; *Barnett, Ward & Anderson, 2021*; *Thorson et al., 2023*). Models with SVCs can be thought of as an extension of simple linear regression, where the estimated field $\omega$ represents an estimated spatial intercept and the field $\varsigma$ represents a spatially varying slope; for this case study, we are interested in long term temporal trends. We can extend the notation in Case Study 1 to represent this model as

$$\boldsymbol{u}_t = f^{-1}\big(\mathbf{Xb} + \boldsymbol{\omega} + \mathbf{X}^{\text{SVC}}\varsigma\big)$$

We use $\mathbf{X}^{SVC}$ as a design matrix for variables associated with the spatially varying coefficient (here $\mathbf{X}^{SVC}$ just includes a continuous vector corresponding to years). When the covariate of interest is time, values of the intercept field $\omega$ that are below average correspond to fine scale locations with consistently below average biomass density, and similarly, values of the slope field $\varsigma$ that are below average can be used to identify fine scale locations where average annual trends in biomass density are declining. Such approaches may be particularly useful in situations where population trends are patchy or exhibit spatial gradients within a large domain (*Davidson et al., 2025*).

To construct the coastwide model of sablefish, we combined trawl survey data (2003–2023) from three regions in the *surveyjoin* package: the West Coast of the U.S., British Columbia, and the Gulf of Alaska. We used a cutoff distance of 25 km to construct the SPDE mesh ($n$ = 944 vertices). Total CPUE was modeled using a Poisson-link delta-lognormal family (*Thorson, 2018*). We followed the same approaches in previous case studies to evaluate model convergence and diagnostics.

We then used the fitted model to predict on to the grids associated with each survey. Since a Poisson-link function was used, the main and spatial effects from each linear predictor were on the same scale (log) and could be summed into a single intercept field. The main and spatially varying coefficient effects of year were similarly summed into a single slope field. Estimates from the coastwide model of sablefish demonstrate that the West Coast of the U.S. and offshore regions in the Gulf of Alaska appear to consistently have above average biomass density for sablefish, while waters in British Columbia appear slightly below average (Fig. 4). With the exception of southern California, estimates of biomass were trending positively for nearly all of the coastline (strongest positive trends occurred in British Columbia and the Gulf of Alaska; Fig. 4).

## DISCUSSION

Managing fisheries across political borders presents challenges that stem from governance, ecological, and logistical complexities. Many fish stocks move across exclusive economic zones (EEZs), requiring coordinated scientific assessments and management strategies. Even within single countries, species shifts necessitate cooperation among stakeholders across management units (*Dubik et al., 2019*). Combining data across jurisdictions can help address these challenges; however, differences in data collection methods, priorities, and governance structures create barriers to effective collaboration. The *surveyjoin* R package represents an initial step toward addressing these challenges by providing a standardized, publicly available database that integrates bottom trawl survey data from

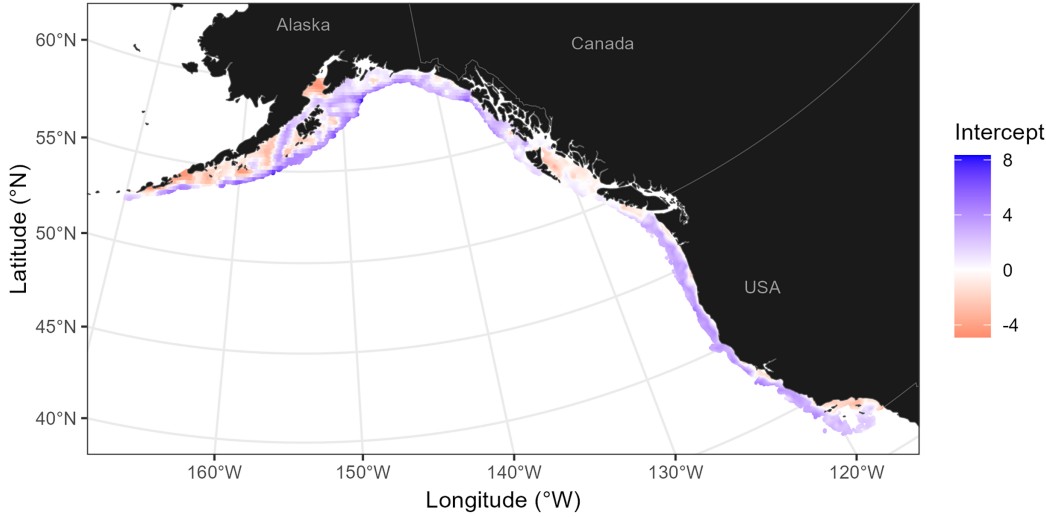

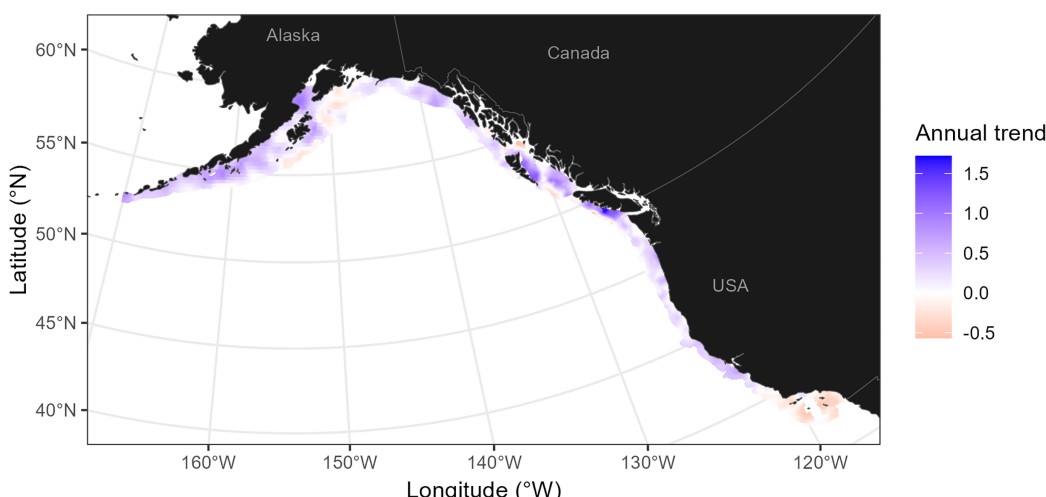

**Figure 4 Estimated intercept and trend for sablefish in the U.S. and Canada.** Estimated intercept (top row) and spatially varying trend (bottom) for sablefish, using trawl survey data 2003–2023. Locations on the intercept spatial field with above average biomass density are indicated with blue, and regions where biomass density is increasing faster than average are highlighted with blue trends. Maps are generated using data from the maps R package (*Becker et al., 2025*) and data from individual surveys (Table 1).

NOAA Fisheries and Fisheries and Oceans Canada. This effort establishes a valuable tool for researchers and managers working across jurisdictions. While similar data standardization efforts exist for ICES-managed fish populations in the North Atlantic, our effort represents one of the first efforts of its kind in the Northeast Pacific.

As some commercially important species are shifting their distribution in response to environmental variation (*Dubik et al., 2019*), the need for data integration across boundaries has become increasingly important. Traditional fisheries management frameworks, which often rely only on jurisdiction-specific data, may not fully capture

large-scale spatial shifts and may confound changes in productivity and movement without this broader context. The ability to synthesize survey data across the full geographic range of a species allows for improved detection of distributional changes and more responsive management. As illustrated by our first case study on Pacific hake, indices generated from the trawl survey provide evidence of different trends in Canadian and U.S. waters over the last 5 years, with an uptick in hake biomass in the southern portion of its range. These relative trends indicate that the center of gravity of hake distribution has shifted south in recent years, though the mechanism responsible for this change is unknown. Anecdotally, this shift is also consistent with the most recent hake population assessment, showing fisheries catches in Canada over the last 2 years have been below expectations (*Johnson et al., 2025*). The exact mechanisms responsible for discrepancies between the acoustic and trawl indices are also unknown, but may be related to gear selectivity, or differences in the seasonal timing of these surveys.

Joining data across political boundaries also offers statistical advantages. A key challenge in geostatistical modeling is the treatment of edge effects and boundaries (*Dahlhaus & Künsch, 1987*). Edge effects arise because most geostatistical models, including the kinds of models used in our case studies, often assume smooth and stationary spatial processes across a study area (*Cressie, 1993*; *Lindgren, Rue & Lindström, 2011*). Estimates near borders may be affected by biases and associated with greater uncertainty (*Heaton et al., 2019*). Implementing boundaries around SPDE meshes represents one approach to alleviating these issues, however another solution is to add additional data, increasing the overall spatial domain. In practical terms, this means that adding data from British Columbia may reduce potential edge effects when predicting to the northern parts of Washington state, or southern portion of the Gulf of Alaska.

By consolidating trawl survey data from NOAA Fisheries and Fisheries and Oceans Canada into a single publicly available R package, *surveyjoin* provides a model for how open science and data initiatives can be expanded to enhance collaboration between agencies and researchers. Beyond its applications for single-species fisheries stock assessments, *surveyjoin* also provides a foundation for broader ecosystem-based management. As bottom trawl surveys collect data on a range of species, these standardized datasets may be used to assess spatial and temporal trends in community structure and biodiversity. Additionally, these data may also be used by industry stakeholders or government partners, including tribal organizations, K-12 classrooms, academic institutions, and non-profit groups.

This dataset could be enhanced in the future in several ways to increase the impact on fisheries science and management. First, the number of species could be increased from the 55 currently included to all species encountered (>1,000). Second, the data backend could be modified to be a remotely hosted database (rather than a database within an R package) for better scalability. Third, the database could be expanded to incorporate additional biological data associated with individual fish, such as age, size, and reproductive traits, which would allow for coastwide analyses of life-history variation and recruitment patterns. Fourth, this database could be expanded to include a broader suite of environmental and habitat variables (particularly for application to dynamic essential fish

habitat analyses). Finally, integrating other survey types, including longline, near-shore, and acoustic surveys, would provide complementary data sources for assessing fish stock distributions across habitat types.

## CONCLUSIONS

The success of fisheries management across borders depends on the ability of scientists, managers, and policymakers to access and interpret data that reflect the full spatial range of fish populations. By bridging gaps between separate monitoring programs and enabling cross-border analyses, *surveyjoin* represents an important step toward more effective, data-driven fisheries management. Our database represents a template that could be easily extended to other regions in North America, or elsewhere around the world. Moving forward, continued collaboration between agencies and international partners will be essential for refining and expanding this resource. Investments in data standardization, open science practices, and interdisciplinary research partnerships will help ensure that fisheries science remains responsive to emerging challenges.

## ACKNOWLEDGEMENTS

The authors thank Lauren Rogers for helpful comments that improved the quality and clarity of our manuscript, to Derek Bolser and Melissa Karp for feedback, and to the joint Fisheries and Oceans Canada–National Oceanic Atmospheric Administration Pacific working group for inspiring this project. We also thank the survey teams and data support staff at Fisheries and Oceans Canada and NOAA Fisheries for collecting and maintaining these long running datasets.

### Funding
The authors received no funding for this work.

### Competing Interests
Eric J. Ward is an Academic Editor for PeerJ.

### Author Contributions
- Eric J. Ward conceived and designed the experiments, performed the experiments, analyzed the data, prepared figures and/or tables, authored or reviewed drafts of the article, and approved the final draft.
- Philina A. English conceived and designed the experiments, performed the experiments, analyzed the data, prepared figures and/or tables, authored or reviewed drafts of the article, and approved the final draft.
- Christopher N. Rooper conceived and designed the experiments, performed the experiments, analyzed the data, prepared figures and/or tables, authored or reviewed drafts of the article, and approved the final draft.

- Bridget E. Ferriss conceived and designed the experiments, performed the experiments, analyzed the data, prepared figures and/or tables, authored or reviewed drafts of the article, and approved the final draft.
- Curt E. Whitmire conceived and designed the experiments, performed the experiments, analyzed the data, prepared figures and/or tables, authored or reviewed drafts of the article, and approved the final draft.
- Chantel R. Wetzel conceived and designed the experiments, performed the experiments, analyzed the data, prepared figures and/or tables, authored or reviewed drafts of the article, and approved the final draft.
- Lewis A. K. Barnett conceived and designed the experiments, performed the experiments, analyzed the data, prepared figures and/or tables, authored or reviewed drafts of the article, and approved the final draft.
- Sean C. Anderson conceived and designed the experiments, performed the experiments, analyzed the data, prepared figures and/or tables, authored or reviewed drafts of the article, and approved the final draft.
- James T. Thorson conceived and designed the experiments, performed the experiments, analyzed the data, prepared figures and/or tables, authored or reviewed drafts of the article, and approved the final draft.
- Kelli F. Johnson conceived and designed the experiments, performed the experiments, analyzed the data, prepared figures and/or tables, authored or reviewed drafts of the article, and approved the final draft.
- Julia Indivero conceived and designed the experiments, performed the experiments, analyzed the data, prepared figures and/or tables, authored or reviewed drafts of the article, and approved the final draft.
- Emily H. Markowitz conceived and designed the experiments, performed the experiments, analyzed the data, prepared figures and/or tables, authored or reviewed drafts of the article, and approved the final draft.

## Data Availability

The data and package are available at GitHub and Zenodo:

- https://github.com/DFO-NOAA-Pacific/surveyjoin-paper.

- Ward, E., Barnett, L., Anderson, S., Johnson, K., Markowitz, E., Ferriss, B., Rooper, C., English, P., Thorson, J., Wetzel, C., Whitmire, C., Fellows, J., & Indivero, J. (2025). DFO-NOAA-Pacific/surveyjoin: v0.0.4 (v0.0.3). Zenodo. https://doi.org/10.5281/zenodo.14984411.

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
