# Peer review of "surveyjoin: a standardized database of scientific trawl surveys in the Northeast Pacific Ocean"

_PeerJ, doi:10.7717/peerj.19964_

## Round 0.1 · original submission · Minor Revisions

Dear Authors, thank you very much for your manuscript. As you can see, all the reviewers agree that your MS is good enough to and written well to have just a minor revision. I personally found it very clear and Interesting. Please read the comments of the reviewers and answer their questions. Following some personal comments.

Lines 98-107: The article you cited (Maureaud et al., 2024) shows the procedure to standardize the different types of data collected, assessing also that users can update the dataset even from other regions of the world. Could your package also be enlarged with other surveys programs apart from those already loaded? Can you give a flowchart showing the steps to standardise the surveys loaded into the package? This could give to the end user the idea of how your rationale worked. Which is the main difference between your database and,e.g., FISHGLOB_data? Why yours could be better or different?

Line 153: Can you provide a formal formula for CPUE? Just to have an idea of how it was calculated across all surveys, since as you perfectly know CPUE could be calculated both on a spatial basis and on a temporal one. Is there any possibility for the end user to calculate CPUE by itself?

Line 190: Did you foresee for an automated update for scientific names from, e.g., Worms portal or similar? Also, the description of species selection criteria (lines 180-190) could be further explained, for example with a flow chart for the final selection of the 55 species.

The case studies showed perfectly the application of the database and, since the subject of the MS is not centered on the case studies I won’t give any other comments on them.

Figure 1: Please, can you enlarge the names of the seas and of the islands? Thank you.

Reviewer 1 ·

Basic reporting

No comment

Experimental design

No comment

Validity of the findings

No comment

Additional comments

Thank you for sharing this excellent work. The manuscript is well-written and thoughtfully presented. Below are a few suggestions for minor revisions to enhance clarity, consistency, and overall polish.

Figures
• To improve clarity and consistency, consider including figure numbers in all figure labels. As seen with Figure 2, this aids navigation and makes it easier for readers to reference specific figures throughout the manuscript.
• The manuscript clearly explains the acoustic survey and its variability, and Figure 2 effectively reflects this by omitting a confidence interval for the acoustic biomass estimate. However, the legend implies both time series (acoustic and coastwide trawl) include confidence intervals, while only one does. For clarity, consider revising the legend or adding a note in the caption to explain this distinction.

Formatting
• In the caption of Table 1, should the term ‘surveyjoin’ be italicized for consistency with its formatting elsewhere in the manuscript? Ensuring uniform formatting throughout will help maintain clarity and a polished presentation.
• A missing period '.' , after the initial "R" in "Tibshirani, R." (line 487) and "H" in Lindgren, F., and Rue, H. (line 521) to align with standard reference formatting. This small fix contributes to the overall polish of the reference list.

References
• Great attention to detail in citing recent work. I noticed a small discrepancy between the in-text citation ('Anderson et al., 2024', lines 241–242) and the reference list (2025, line 445). Aligning the publication year would enhance consistency and clarity.
• The reference to Lindgren, F., and Rue, H. (2015) on line 521 is a valuable source (not cited). To ensure completeness, consider integrating a corresponding in-text citation if it was intended to be referenced.

Reviewer 2 ·

Basic reporting

This manuscript presents an R package that provides open access to a standardized database of bottom trawl survey data collected in the Northeast Pacific Ocean by NOAA Fisheries and Fisheries and Oceans Canada. The manuscript is well written, clearly organized, and of strong relevance to researchers working on fisheries sciences and marine conservation. It contains clear, well-labeled figures and tables that nicely illustrate the main findings. All of the raw survey data are made available in the package, ensuring full reproducibility. However, the manuscript would benefit from the inclusion of a few more references to support key statements (see minor comments below). By enabling access to standardized, transboundary survey data, the package will support research on shared (straddling) stocks and contribute to more informed management decisions across US/Canada boundaries.

Experimental design

The manuscript clearly frames its primary research question, how to standardize and leverage bottom-trawl data to inform transboundary stocks management, particularly through the use of spatio-temporal ecological modeling, and demonstrates how this fills a key gap in cross-jurisdictional stock assessment workflows. The design and structure of the manuscript are solid. The authors presented the new R package and describe the data sources and how standardization was done. It also includes three practical examples illustrating how the database can be used to address ecological questions, particularly using spatio-temporal models with the tinyVAST and sdmTMB R packages. The code is clearly implemented, publicly available via GitHub, and reproducible, which enhances the utility and transparency of the work.

Validity of the findings

The primary contribution of this manuscript is the development of a standardized and accessible R package and accompanying dataset, offering a novel tool that fills a gap in transboundary stock distribution and will likely become a cornerstone resource for fisheries scientists. This contribution is timely and valuable, and the examples provided demonstrate its practical application. All underlying survey data are available via the GitHub repository; they have been quality-checked, standardized, and subjected to validation to ensure robustness and proper control.

Additional comments

I thoroughly enjoyed reading this manuscript. Open-access datasets like the one described are needed for advancing scientific understanding and promoting collaborative research. As the authors correctly note, standardized fisheries data are essential for the sustainable management of shared fish stocks, especially in the context of climate change and shifting species distributions.
I commend the authors for their efforts in creating and documenting this resource, which will be highly beneficial for researchers working on fisheries and ecosystem modeling in the Pacific Northwest. The manuscript is professionally written and generally well supported, though there are a few areas where additional references would strengthen the text.

Minor Comments:
• Lines 64–65: The current wording may unintentionally suggest that nearly half of all coastal states’ catch comes from shared stocks. Consider rephrasing for clarity, e.g.:
“For countries that participate in shared-stock fisheries, these stocks represent nearly half of their total catch (Teh & Sumaila, 2015).”
• Line 74: Consider adding this relevant reference:
Gaines, S.D. et al. (2018). Improved fisheries management could offset many negative effects of climate change. Sci. Adv. 4, eaao1378. https://doi.org/10.1126/sciadv.aao1378
• Line 168: Please provide a reference to support the statement about borealization. Suggested citation:
Litzow, M.A., Fedewa, E.J., Malick, M.J. et al. (2024). Human-induced borealization leads to the collapse of Bering Sea snow crab. Nat. Clim. Chang. 14, 932–935. https://doi.org/10.1038/s41558-024-02093-0
• Lines 226–227: Please clarify whether the trawl survey primarily captures adult Pacific hake, or if juveniles are also well represented.
• Lines 270–271: The discrepancy between the acoustic and bottom trawl indices is striking. A brief explanation or hypothesis for these differences would help readers better understand the limitations and interpretation of each method.
• Line 277: Please add a reference to support the statement about the biogeographic break in the California Current. A possible source might be one discussing community structure or oceanographic boundaries.
• Lines 283–285: This section would benefit from a supporting citation. One option:
Howard, R.A., Ciannelli, L., Wakefield, W.W., & Fewings, M.R. (2021). The effects of climate, oceanography, and habitat on the distribution and abundance of northern California Current continental shelf groundfishes. Fisheries Oceanography, 30(6), 707–725.

Reviewer 3 ·

Basic reporting

The dataset presented in the manuscript “surveyjoin: a standardized database of scientific trawl surveys in the Northeast Pacific Ocean” is a great contribution to facilitate access to data from scientific bottom trawl surveys in the Northwest Pacific Coast. The companion R package follows the best standard for package development and is well documented. The three examples provided are good illustrations of how to make advanced statistical analysis on bottom trawl surveys from multiple data sources. Overall, the manuscript is of high quality. It was a pleasure to read and review it.
My main concern is about the long-term vision of this new effort at joining bottom trawl surveys in a single platform. Already existing platforms on bottom trawl surveys (OceanAdapt, Fishglob) lack recent updates (the latest data from the Northwest Pacific Coast is from 2019). Yet, these two existing platforms offer the pre-processing scripts that can be used to clean and harmonize the newly released data. Furthermore, in the case of Fishglob, they continuously encourage new members to join their on-going effort (see Maureaud et al. 2025, doi: 10.1111/csp2.70035). While I can see the value of a regional data gathering effort, I am not sure whether this new initiative will lead to a continuously updated dataset. In this perspective, I strongly encourage the authors to share their data pre-processing scripts to facilitate future uses of their data pipeline and ensure consistency along the time series and among databases.

Experimental design

The manuscript is a data paper, so it doesn’t include a proper research question nor an experimental design. The dataset is coherent and fills a need for fisheries management. It is well presented in Figure 1 and Table 1. Issues with joining data from multiple sources are well discussed. Differences in the survey protocol (gears, speed, duration) may lead to differences in catchability. While the authors mentioned that catchability bias can be estimated statistically, none of the three illustrative case studies consider ‘surveys’ or ‘gear’ as a variable. I would be curious to see whether the random effect of ‘survey’ is small and negligible. This information could be important for future users of the database.
Another issue that typically arises when working with multiple long-term scientific surveys is the inconsistency in species identification. I would encourage the authors to describe more in details how they handled this issue, e.g. some taxa identified at genus level, or with known mis-identification errors.
One last minor remark: Raja binoculata is not an accepted taxa anymore in Fishbase (it is now called Beringraja binoculata) which might lead to compatibility issues when working with other data sources. However, it is true that ITIS does not recognize the new genus Beringraja yet.

Validity of the findings

As a data paper, I did not judge the quality of the three case studies which are only illustrative. Merging and standardizing databases of scientific trawl surveys have clear benefits for fisheries scientists. Yet I am confused about the possible biases emerging from joining datasets with inconsistent sampling strategy, leading to temporally inconsistent spatial coverage. What triggered my curiosity was the strong decrease in biomass of Pacific hake in 2020 (Figure 2), which is mostly due to the lack of data during Covid pandemic. While this effect can be easily spotted because most surveys were missing in 2020, it is less clear what is the effect of bi-annual or tri-annual surveys on the estimated spatio-temporal dynamics of species. In this first case study, the period 2003-2023 is not consistently sampled: the DFO survey alternates between Hecate Strait and Queen Charlotte Sound in odd years; Haida Gwaii and Vancouver Island in even years (as explained in Table 1). If the hake is not equally distributed in the area, then the annual density will be biased depending on the spatial coverage of the surveys in a given year. This would call for the use of a time-windows or a decrease in the temporal resolution.
This is a critical aspect of the surveyjoin dataset because only two of the fourteen surveys are annual (eastern Bering Sea and NWFSC.Combo). Therefore, every year has a different spatial coverage. I would encourage the authors to discuss this limitation and its methodological considerations in the manuscript.

Additional comments

I don't have specific additional comments. The manuscript was clear and well written.

---

## Round 0.2 · accepted · Accept

Dear authors, The reviewers and I are satisfied with the corrections made to the manuscript. No other concerns emerged, so the final decision is to accept it for publication.

Reviewer 1 ·

Basic reporting

no comment

Experimental design

no comment

Validity of the findings

no comment

Additional comments

Thank you for submitting the revised manuscript. I appreciate the thoughtful effort you've put into addressing the reviewer comments. The paper presents a meaningful contribution to marine ecological studies, and the software package developed is both timely and highly relevant for ecological applications. The revisions have strengthened the manuscript. I believe this work will be of interest to many in the field.

Reviewer 2 ·

Basic reporting

The manuscript remains clear, well-organized, and relevant. The authors have addressed the suggestion to include additional references to support key statements. Specifically, they clarified the language regarding shared-stock catch (Lines 65–66), and incorporated the suggested citations (e.g., Gaines et al. 2018; Litzow et al. 2024; Howard et al. 2021), which strengthens the contextual framing. Figures and tables continue to be clear and informative. The open-access data and reproducibility of the R package are maintained and well documented.

Experimental design

The authors continue to present a clear and well-defined objective. The manuscript describes the R package structure, data standardization process, and utility through practical examples. No changes were required in this section, but the authors ensured the original clarity and structure were preserved. The examples using tinyVAST and sdmTMB remain appropriate and illustrative, and all code remains available and reproducible via GitHub.

Validity of the findings

The authors’ revisions do not alter the fundamental contribution of the manuscript, which remains a valuable and timely resource for transboundary fisheries science. They addressed specific clarifications requested (e.g., clarified whether Pacific hake juveniles are represented in the survey [Lines 230--231], and provided a brief explanation for the discrepancies between survey indices [Lines 403-405]). These additions enhance the interpretability and transparency of the data resource.

Additional comments

The authors have thoughtfully addressed all minor comments and incorporated the recommended changes. The resulting manuscript is strengthened through improved clarity, added citations, and helpful clarifications. I commend the authors for their responsiveness and continue to view this work as a valuable contribution to the field. I recommend the manuscript be accepted.

Reviewer 3 ·

Basic reporting

I want to acknowledge the authors for answering in detail all the comments. I think that the manuscript has improved substantially. Therefore, I recommend the manuscript for publication. It is a great contribution to facilitate access to harmonized scientific bottom trawl survey datasets across boundaries in the Northwest Pacific Coast.

Experimental design

no comment

Validity of the findings

I am not convinced by the explanation about the drop in biomass estimate in 2020 (Figure 2), which is an artifact of the low number of samplings. But this is only an illustrative example (and not a stock assessment), so it doesn’t influence the relevance and the quality of the database.

Additional comments

I have no further comment.